# Combined genome-wide association study of 136 quantitative ear morphology traits in multiple populations reveal 8 novel loci

Yi Li[1,2☯], Ziyi Xiong[3,4☯], Manfei Zhang[1,5,6☯], Pirro G. Hysi[7☯], Yu Qian[2,8☯], Kaustubh Adhikari[9,10☯], Jun Weng[11], Sijie Wu[1,6,12], Siyuan Du[1,5,13], Rolando Gonzalez-Jose[14], Lavinia Schuler-Faccini[15], Maria-Catira Bortolini[15], Victor Acuna-Alonzo[16], Samuel Canizales-Quinteros[17], Carla Gallo[18], Giovanni Poletti[18], Gabriel Bedoya[19†], Francisco Rothhammer[20], Jiucun Wang[6,12], Jingze Tan[12], Ziyu Yuan[21], Li Jin[1,6,12,21], André G. Uitterlinden[4,22], Mohsen Ghanbari[4], M. Arfan Ikram[4], Tamar Nijsten[23], Xiangyu Zhu[11,24], Zhen Lei[11,24], Peilin Jia[2], Andres Ruiz-Linares[9,12,25‡], Timothy D. Spector[7‡], Sijia Wang[1,26‡*], Manfred Kayser[3‡], Fan Liu[2,3‡*]

1 CAS Key Laboratory of Computational Biology, Shanghai Institute of Nutrition and Health, Chinese Academy of Sciences, China, 2 CAS Key Laboratory of Genomic and Precision Medicine, Beijing Institute of Genomics, Chinese Academy of Sciences, China, 3 Department of Genetic Identification, Erasmus MC, University Medical Center, the Netherlands, 4 Department of Epidemiology, Erasmus MC, University Medical Center, the Netherlands, 5 Bio-X Institutes, Key Laboratory for the Genetics of Developmental and Neuropsychiatric Disorders, Ministry of Education, Shanghai Jiao Tong University, Shanghai, China, 6 State Key Laboratory of Genetic Engineering, Human Phenome Institute, Zhangjiang Fudan International Innovation Center, Fudan University, China, 7 Department of Twin Research and Genetic Epidemiology, King's College London, United Kingdom, 8 Beijing No.8 High School, Beijing, China, 9 Department of Genetics, Evolution and Environment, and UCL Genetics Institute, University College London, United Kingdom, 10 School of Mathematics and Statistics, Faculty of Science, Technology, Engineering and Mathematics, The Open University, United Kingdom, 11 Center for Biometrics and Security Research & National Laboratory of Pattern Recognition, Institute of Automation, Chinese Academy of Sciences, China, 12 Ministry of Education Key Laboratory of Contemporary Anthropology, Collaborative Innovation Center for Genetics and Development, School of Life Sciences, Fudan University, China, 13 University of Chinese Academy of Sciences, China, 14 Instituto Patagonico de Ciencias Sociales y Humanas, Centro Nacional Patagonico, CONICET, Argentina, 15 Departamento de Genetica, Universidade Federal do Rio Grande do Sul, Brasil, 16 Molecular Genetics Laboratory, National School of Anthropology and History, Mexico, 17 Unidad de Genomica de Poblaciones Aplicada a la Salud, Facultad de Quimica, UNAM-Instituto Nacional de Medicina Genomica, Mexico, 18 Laboratorios de Investigacion y Desarrollo, Facultad de Ciencias y Filosofía, Universidad Peruana Cayetano Heredia, Peru, 19 GENMOL (Genetica Molecular), Universidad de Antioquia, Medellin, Colombia, 20 Instituto de Alta Investigacion, Universidad de Tarapaca, Chile, 21 Fudan-Taizhou Institute of Health Sciences, China, 22 Department of Internal Medicine, Erasmus MC, University Medical Center, the Netherlands, 23 Department of Dermatology, Erasmus MC, University Medical Center, the Netherlands, 24 School of Artificial Intelligence, University of Chinese Academy of Sciences, China, 25 Aix-Marseille Universite, CNRS, EFS, ADES, France, 26 Center for Excellence in Animal Evolution and Genetics, Chinese Academy of Sciences, China

☯ These authors contributed equally to this work.
† Deceased.
‡ AR-L, TDS, SW, MK and FL also contributed equally to this work.
* wangsijia@picb.ac.cn (SW); f.liu@erasmusmc.nl (FL)

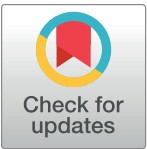

**Data Availability Statement:** All relevant data to run future replications and meta-analysis efforts are provided online (https://doi.org/10.6084/m9.figshare.20961640.v1). This includes the summary

## Abstract

Human ear morphology, a complex anatomical structure represented by a multidimensional set of correlated and heritable phenotypes, has a poorly understood genetic architecture. In this study, we quantitatively assessed 136 ear morphology traits using deep learning analysis of digital face images in 14,921 individuals from five different cohorts in Europe, Asia,

statistics for 136 meta-analysis, the adjusted minimal p values of 136 meta-analysis and the p values from C-GWAS, for all used SNPs (~4 million). The participants involved in The Rotterdam Study (RS), The TwinsUK Study (TwinsUK), The Taizhou Longitudinal Study (TZL), The National Survey of Physical Traits Study (NSPT) and The Consortium for the Analysis of the Diversity and Evolution of Latin America (CANDELA) Study datasets were not collected with broad data sharing consent. Given the identifiable nature of both ear and genomic information and unresolved issues regarding risks to participants of inherent reidentification, participants were not consented for inclusion in public repositories or the posting of individual data, meaning that the raw ear images and/or individual-level genomic data cannot be made publicly available due to restrictions imposed by the ethics approval. The same phenotype acquisition pipeline used by different cohorts is available at Github (https://github.com/Fun-Gene/EarPhenotyping).

**Funding:** Specified authors were supported by research grants as follows: Strategic Priority Research Program of Chinese Academy of Sciences, XDB38010400 (FL), XDB38020400 (SW), and XDC01000000 (FL); Shanghai Municipal Science and Technology Major Project (2017SHZDZX01 to FL, SW, and LJ); the National Natural Science Foundation of China (NSFC) 81930056 (FL), 31521003 (SW), and 31900408 (MZ); Science and Technology Service Network Initiative of Chinese Academy of Sciences KFJ-STS-ZDTP-079 (FL); the Beijing Advanced Discipline Fund (FL); CAS Interdisciplinary Innovation Team Project (SW); Shanghai Science and Technology Commission Excellent Academic Leaders Program 22XD1424700 (SW); the CAS Project for Young Scientists in Basic Research YSBR-077 (SW); National Science & Technology Basic Research Project 2015FY111700 (LJ); the 111 Project B13016 (LJ); China Postdoctoral Science Foundation 2019M651352 (MZ); Leverhulme Trust grant F/07 134/DF (A.R.-L.); Biotechnology and Biological Sciences Research Council grant BB/I021213/1 (A.R.-L.). The Rotterdam Study is supported by the Erasmus MC and Erasmus University Rotterdam; the Netherlands Organization for Scientific Research (NWO); the Netherlands Organization for Health Research and Development (ZonMw); the Research Institute for Diseases in the Elderly (RIDE) the Netherlands Genomics Initiative (NGI); the Ministry of Education, Culture and Science; the Ministry of Health Welfare and Sport; the European Commission (DG XII); and the Municipality of Rotterdam. The generation and management of

and Latin America. Through GWAS meta-analysis and C-GWASs, a recently introduced method to effectively combine GWASs of many traits, we identified 16 genetic loci involved in various ear phenotypes, eight of which have not been previously associated with human ear features. Our findings suggest that ear morphology shares genetic determinants with other surface ectoderm-derived traits such as facial variation, mono eyebrow, and male pattern baldness. Our results enhance the genetic understanding of human ear morphology and shed light on the shared genetic contributors of different surface ectoderm-derived phenotypes. Additionally, gene editing experiments in mice have demonstrated that knocking out the newly ear-associated gene (*Intu*) and a previously ear-associated gene (*Tbx15*) causes deviating mouse ear morphology.

## Author summary

Human ear morphology varies widely, with largely inherited differences. However, the genetic basis for these differences is not well understood. To better understand the genetics of human ear morphology, we analyzed 136 ear traits from digital facial images of 14,921 individuals from three continents using GWAS meta-analysis and the recently developed C-GWAS method. We identified 16 genetic loci associated with ear morphology, eight of which had not been previously associated with ear features. These genetic loci can serve as candidates for future research to understand the function of these genes. Additionally, we have demonstrated the impact of two ear-associated loci on ear morphology in gene-edited mice. This study provides new insights into the complex genetics of the human ear.

## Introduction

Human ear morphology is a complex trait that is influenced by a multidimensional set of correlated and heritable phenotypes. Studies have estimated the heritability of ear morphology phenotypes to be between 29% and 61% [1], indicating that genetic variants play a significant role in shaping the ear's anatomy. Understanding the genetic basis of human ear morphology has important implications for multiple scientific disciplines such as human genetics, developmental biology, medicine, evolutionary biology, anthropology, and forensics. This study aims to generate knowledge that can aid in understanding genetic disorders related to ear morphology, improve diagnosis and treatment of ear-related diseases and also help in forensics and anthropology to identify individuals based on their ear morphology.

Previous genome-wide association studies (GWASs) on ear morphology have focused on a limited number of qualitative ear features and have not utilized quantitative ear traits. A GWAS in 5,062 Latin Americans identified significant associations at seven genetic loci for ear lobe size and attachment, folding of antihelix, helix rolling, ear protrusion and antitragus size, suggesting the involvement of the genes *EDAR*, *CART1*, *SP5*, *MPRS22*, *LRBA*, *LOC153910*, and *LOC100287225* [1]. A multi-ethnic GWAS meta-analysis on earlobe attachment in 74,660 samples revealed a total of 49 genetic loci [2], further emphasizing the complexity of the genetic architecture of ear morphology.

Due to the complex and multidimensional nature of ear morphology, focusing on a limited number of qualitative ear features, as done in previous GWASs, can result in missing

GWAS genotype data for the Rotterdam Study were executed by the Human Genotyping Facility of the Genetic Laboratory for Population Genomics of the Department of Internal Medicine, Erasmus MC. TwinsUK is funded by the Wellcome Trust, Medical Research Council, Versus Arthritis, European Union Horizon 2020, Chronic Disease Research Foundation (CDRF), Zoe Ltd and the National Institute for Health Research (NIHR) Clinical Research Network (CRN) and Biomedical Research Centre based at Guy's and St Thomas' NHS Foundation Trust in partnership with King's College London. P.G.H acknowledges the use of funding from the BrightFocus Foundation. The generation and management of GWAS genotype data for the TZL and NSPT was collected and executed by the collaboration of CAS Key Laboratory of Computational Biology, Shanghai Institute of Nutrition and Health, Chinese Academy of Sciences, Shanghai, China and the State Key Laboratory of Genetic Engineering and Ministry of Education Key Laboratory of Contemporary Anthropology, Collaborative Innovation Center for Genetics and Development, School of Life Sciences, Fudan University, Shanghai, China. The funders had no role in study design, data collection and analysis, decision to publish, or preparation of the manuscript.

**Competing interests:** The authors have declared that no competing interests exist.

important phenotypic information and genetic associations. Additionally, traditional methods such as human perception [1] and questionnaire-based [2] phenotyping are prone to bias and instability. Therefore, in this study, we developed a fully automated computer pipeline that utilizes a deep learning convolutional neural network (CNN) [3] to quantify a large number of ear phenotypes from high-resolution digital side-face photos. This approach allows for a more comprehensive and accurate assessment of ear morphology and can provide a more complete understanding of the genetic factors underlying this trait.

High correlations between various ear traits may be influenced by genetic variants with multi-trait effects. However, traditional single-trait GWAS, as used in previous ear GWASs [1,2], has limitations in identifying these genetic loci. To overcome these limitations, here we use C-GWAS [4], a recently developed method for combining GWAS summary statistics of multiple potentially related traits, to integrate our GWAS results. C-GWAS has been shown to have increased statistical power compared to traditional methods such as minimal p-values of multiple single-trait GWASs (MinGWAS) and MTAG [5]. Furthermore, it has been successfully applied to the analysis of a large number of facial traits, resulting in the identification of novel genetic loci associated with facial variation that were not identified via traditional methods [4].

In this study, we used a sample of 14,921 multi-ethnic individuals from Europe ($N$ = 4,740), Asia ($N$ = 4,835), and Latin America ($N$ = 5,346) to investigate the genetic factors that contribute to ear morphology. We quantified 136 quantitative ear phenotypes using deep learning analyses of digital face images, and conducted GWASs and GWAS meta-analyses. These results were then combined using a recently developed C-GWAS method [4]. We identified novel genetic loci and confirmed previously established loci that are involved in normal-range quantitative variation of ear morphology in humans. We performed functional follow-up studies in gene-edited mice to further understand the impact of these genetic loci on ear morphology.

## Results

### Performance of CNN in quantitative ear phenotyping

In this study, we used a deep learning convolution neural network (CNN) approach [3] for comprehensive ear landmarking on high-resolution 2D digital side face photos. We focused on the 17 most anatomically meaningful landmarks (S1A Fig) out of the 55 that could be located by the CNN, and derived 136 pairwise inter-landmark distances as input phenotypes for subsequent genetic analyses. We found that the inter-rater correlations on the left ear were reasonably high (mean r = 0.47, sd = 0.24, S2A Fig) and the correlations between manual- and CNN-derived phenotypes on the same ear were reasonably high too (mean r = 0.34, sd = 0.18, S2B Fig). Utilizing the symmetric nature of the human face, it was obvious that the left and right ear correlations for CNN-derived phenotypes (r = 0.52, sd = 0.10, S2D Fig) were significantly higher, and with a smaller standard deviation, than those from the same human rater (r = 0.44, sd = 0.20, S2C Fig). This demonstrated that the deep learning approach on ear phenotyping that we used in this study on the full set of facial images had at least an equal performance in the accuracy of ear landmarking compared to human perception, while also being more efficient.

### Sample characteristics and phenotype results

This study included 14,921 individuals from five different population cohorts from three continents: Europe, Asia, and Latin America (S1B Fig and S1 Table). These cohorts were: two European cohorts (the Rotterdam Study with 3,675 participants from the Netherlands and the

TwinsUK study with 1,065 participants from the UK), two East Asian cohorts (the Taizhou Longitudinal Study, TZL, with 2,348 Han Chinese participants from China and the National Survey of Physical Traits study, NSPT, with 2,487 Han Chinese participants from China), and one cohort of mixed ancestry (the CANDELA study with 5,346 Latin Americans, who have an estimated genomic admixture of 48% European, 46% Native American, and 6% African). It is worth noting that, while CANDELA and TZL have been used in previous GWAS studies on qualitative ear features [1,2], quantitative ear phenotypes have not been studied in these cohorts before.

The majority (82–100%) of the 136 ear phenotypes studied were normally distributed, as determined by the Kolmogorov-Smirnov normality test ($P > 0.05$ after multiple testing correction, as shown in S2 and S3 Tables). The lower proportion of normally distributed ear phenotypes in the Rotterdam Study (RS) can be attributed to a higher proportion of older individuals (with an average age more than 10 years older than in the other cohorts, as seen in S1 Table). The effects of sex (min $P = 3.5e\text{-}145$) and age (min $P = 5.9e\text{-}99$) on ear phenotypes were investigated in detail using individual-level data from RS. These factors together explained up to 16.9% of the variance for certain ear phenotypes (e.g. phenotype L7-L14) (S4 Table). Women had longer distances involving otobasion inferius (L7) compared to men, while men had longer distances involving otobasion superius (L1) compared to women (S3A Fig). Aging was associated with a significant increase in ear length and thinness (S3B Fig). Unsupervised hierarchical clustering of the 136 ear phenotypes resulted in two main clusters, mainly representing vertical and horizontal ear features (S4A Fig). The genetic correlation estimates from genome-wide SNPs were high and similar to phenotypic correlations (S4B Fig), indicating shared genetic contributors across multiple ear phenotypes. The twin-based heritability of the 136 ear phenotypes was estimated in TwinsUK to have a mean of 0.52 (sd = 0.03, 0.49–0.56, S5A Fig). These estimates were generally higher than those obtained from genome-wide SNPs in the same cohort (mean = 0.38, sd = 0.09, 0.20–0.60, S5B Fig). The genetic heritability estimates were largely consistent with those for qualitative ear features reported previously in the CANDELA study [1].

## GWASs, Meta-analysis, and C-GWAS

Single-trait GWASs of 136 ear phenotypes were conducted in each of the five cohorts separately (as shown in S1B Fig) and the results were then meta-analyzed, resulting in 136 sets of single-trait summary statistics. The inflation factors in these single-trait GWASs were all within an acceptable range (average $\lambda = 1.04$, with $1.02 < \lambda < 1.06$). The minimal p-values of the 136 single-trait GWASs were adjusted empirically using simulations as implemented in C-GWAS, so that the adjusted minimal p-values follow a uniform distribution under the null. These adjusted minimal p-values are abbreviated as "MinGWAS" below. Similarly, the C-GWAS results were also adjusted using the embedded simulations, so that the adjusted p-values also closely follow a uniform distribution under the null (as shown in Fig 1A–1C). This means that MinGWAS and C-GWAS results can be directly compared with each other as well as with those from any standard single-trait GWAS. Therefore, the traditional genome-wide significance threshold ($P \leq 5e\text{-}8$) can be considered as the study-wide significance threshold in this study. Additional information on the C-GWAS method can be found in other publications [4].

MinGWAS and C-GWAS together identified a total of 1,205 ear-associated SNPs at 16 distinct genetic loci that exceeded the study-wide significance (Fig 1D–1E and Table 1). Eight of these loci had not been previously associated with ear features in previous GWASs and were identified for the first time in this study (Fig 1F); four of them were significant in both

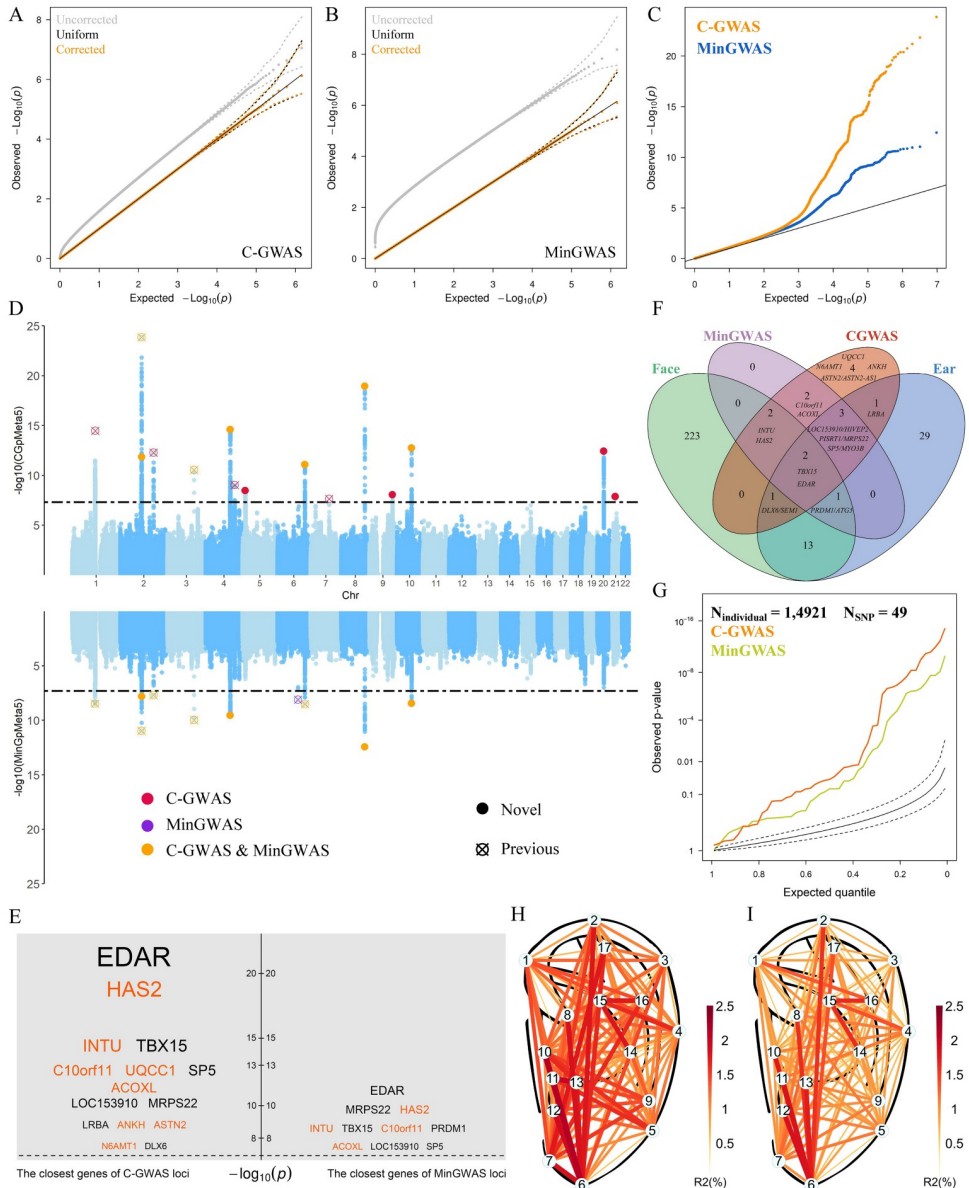

**Fig 1. Outcomes from GWAS meta-analysis (MinGWAS) and combined GWAS (C-GWAS) for 136 quantitative ear morphology traits in 5 multi-ethnic cohorts (N = 14,921).** A simulation analysis derived the null distributions of the crude p-values (gray). The p-values corrected using the getCoef function (orange, details of function in Materials and methods) are compared with the uniform distribution (black) for both the C-GWAS results (**A**) and the MinGWAS result (**B**). After multiple testing correction, the C-GWAS results (upper part of **D**) and the minimal p-values from 136 single ear trait GWASs (lower part of **D**) are plotted using a combined Manhattan plot (**D**) and a Q-Q plot (**C**). In the Manhattan plot (**D**), the study-wide significance threshold (*P* = 5e-8, all p-values of MinGWAS and C-GWAS have been corrected) is indicated using dashed lines. A total of 16 study-wide significant loci were identified, among which 9 were significant in both C-GWAS and MinGWAS (orange dots and circles with cross), 1 solely was significant in MinGWAS (purple circle with cross), and the remaining 6 were significant only in C-GWAS (red dots and circles with cross). (**E**) The closest genes to regional lead SNPs are listed (**E**) (orange for novel). (**F**) Venn diagram of 15 candidate genes in the 15 loci highlighted by C-GWAS, 10 candidate genes in the 10 loci highlighted by MinGWAS, 50 candidate genes in 50 loci from previous GWASs of categorical ear features, and 243 candidate genes in 243 loci from previous GWASs of facial shape variation. (**G**) A list of 59 previously established ear-associated SNPs was looked up in the C-GWAS and MinGWAS results. In RS, the ear variance explained by 15 lead SNPs from loci identified by C-GWAS (**H**) is compared with that explained by 10 lead SNPs from loci identified by MinGWAS (**I**).

MinGWAS and C-GWAS, and four were significant solely in C-GWAS (Table 1 and Fig 1F). Notably, all eight novel loci had highly consistent allele effects across all five cohorts, despite their different continental ancestries (S6A–S6H Fig). The most significant novel finding was rs7812632 at 8q24.13 ($P_{\text{C-GWAS}}$ = 1.2e-19), where the G-allele mainly led to an increased vertical length of the ear. This SNP was top-associated with L6-L15 ($P_{\text{MinGWAS}}$ = 4.7e-10) and showed nominally significant association with L6-L15 in all cohorts, with similar effect sizes (except for TwinsUK due to QC failure). This SNP is located approximately 200kbp upstream of the *HAS2* gene, which encodes hyaluronan synthase 2, a protein that plays an important role in embryonic development of branchial arches [6] and cranial neural crest cells (CNCCs) [7]. Variants in or near *HAS2* have been previously associated with face morphology [8], body height [9], and male pattern baldness [10]. Additionally, this locus is the only one among all the significant ones we found that shows borderline genome-wide significant association with ear landmark L3 ($P_{\text{L3-L15}}$ = 2.7e-7, S6D Fig), which anatomically approximates Darwin's Tubercle (OMIM:124300, S7 Fig) [11]. We also tested the association between rs7812632 and Darwin's Tubercle in CANDELA, as it is the only cohort studied in which this phenotype has been manually obtained. This SNP was nominally associated with Darwin's Tubercle ($P$ = 0.03) and the A-allele was associated with an increased prevalence.

The second most significant novel finding was rs57788627 at 4q28.1 ($P_{\text{C-GWAS}}$ = 2.5e-15), where the G-allele mainly led to an increased ear width. This SNP was top-associated with L15-L16 ($P_{\text{MinGWAS}}$ = 2.3e-9) and showed nominally significant association with L15-L16 in almost all cohorts, with similar effect sizes (4/5, except NSPT). This locus is remarkable as it represents a gene desert and the nearest gene, *INTU*, is 408kb away from the region's lead SNP

**Table 1. SNPs associated with quantitative ear morphology traits identified by MinGWAS and C-GWAS.**

| | | | | | MinGWAS | | C-GWAS | |
| Locus | CHR | Nearby Genes | Novel | SNP | $P$ | Trait | % ($P \leq 0.05$) | $P$ |
|---|---|---|---|---|---|---|---|---|
| 1 | 1p12 | SPAG17;TBX15 | 0 | rs6699106 | **3.43E-09** | L4_L15 | 53.68 | **4.52E-12** |
| | | | | rs17023457 | 7.19E-07 | L10_L13 | 50.74 | **3.46E-15** |
| 2 | 2q12.3 | EDAR | 0 | rs17034666 | **1.09E-11** | L9_L12 | 61.03 | **1.38E-24** |
| 3 | 2q13 | ACOXL | 1 | rs3789101 | **1.57E-08** | L1_L4 | 50.74 | **1.48E-12** |
| 4 | 2q31.1 | SP5;MYO3B;LOC101926913 | 0 | rs6756973 | **2.10E-08** | L6_L7 | 25.74 | **5.21E-12** |
| | | | | rs10198822 | 3.93E-07 | L6_L7 | 30.15 | **5.16E-13** |
| 5 | 3q23 | PISRT1;MRPS22 | 0 | rs10212419 | **1.05E-10** | L6_L15 | 40.44 | **2.85E-11** |
| 6 | 4q28.1 | MIR2054;INTU | 1 | rs57788627 | **2.31E-09** | L15_L16 | 55.88 | **2.54E-15** |
| 7 | 4q31.3 | LRBA | 0 | rs6535728 | 1.27E-05 | L5_L13 | 53.68 | **9.69E-10** |
| 8 | 5p15.1 | ANKH;LINC02149 | 1 | rs10062331 | 5.32E-07 | L11_L16 | 42.65 | **3.46E-09** |
| 9 | 6q21 | PRDM1;ATG5 | 0 | rs61123157 | **7.82E-09** | L6_L11 | 41.91 | 1.30E-06 |
| 10 | 6q24.2 | LOC153910;HIVEP2 | 0 | rs7771119 | **1.98E-08** | L6_L12 | 42.65 | **8.39E-12** |
| 11 | 7q21.3 | SEM1;DLX6-AS1;DLX6 | 0 | rs12535056 | 1.62E-03 | L9_L14 | 33.09 | **2.31E-08** |
| 12 | 8q24.13 | HAS2;HAS2-AS1;SMILR | 1 | rs7812632 | **4.67E-10** | L6_L15 | 55.15 | **1.19E-19** |
| 13 | 9q33.1 | ASTN2;ASTN2-AS1 | 1 | rs4837613 | 2.33E-07 | L11_L13 | 47.79 | **8.70E-09** |
| 14 | 10q22.2 | C10orf11(LRMDA) | 1 | rs10824309 | **4.83E-09** | L6_L12 | 43.38 | **1.82E-13** |
| 15 | 20q11.22 | UQCC1 | 1 | rs2378353 | 3.42E-06 | L6_L13 | 34.56 | **3.87E-13** |
| 16 | 21q21.3 | LINC00161;N6AMT1 | 1 | rs9982925 | 4.24E-05 | L8_L17 | 46.32 | **1.40E-08** |

P value passed the study-wide significant threshold are indicated in bold; The most significant SNP in each locus was selected; Novel means that the locus has not been previously reported for association with ear phenotypes; minP, the minimal corrected p value in the locus; minP-Trait, the phenotype with which the minimal corrected p-value was obtained; % ($P \leq 0.05$), the percentage of phenotypes with p-values $\leq 0.05$; Study-wide significant threshold is 5e-8.

(S6B Fig). A recent face GWAS by Xiong et al. found an association between *INTU* and human facial variation and also confirmed the enhancer activity of this region through luciferase assay experiments [8]. Among all genes in this region, *INTU* showed the most preferential expression in CNCCs. Additionally, *INTU* is involved in both ciliogenesis and convergent extension and plays a role in embryonic development [12]. Recent studies have suggested an important role of *Intu* in mouse skeletal development [13]. Furthermore, exome sequencing of orofaciodigital syndrome patients with facial, oral, and ear abnormalities revealed disease-associated mutations in *INTU* [14]. We attained mouse mutants using one-step CRISPR/cas9 mediated gene editing experiments to assess the functional significance of *Intu* gene expression during ear mouse development (for details, see below).

Among the eight loci that were already reported in the two previous single-trait GWASs of qualitative ear traits, five were significant in our quantitative study in both MinGWAS and C-GWAS, specifically 1p12 *TBX15*, 2q12.3 *EDAR*, 2q31.1 *SP5*, 3q23 *MRPS22*, and 6q24.2 *HIVEP2*. One locus, 6q21 *PRDM1/ATG5*, reached study-wide significance only in MinGWAS, and two, 4q31.3 *LRBA* and 7q21.3 *DLX6*, reached study-wide significance only in C-GWAS. The allele effects of all these eight loci were in the same direction in all five cohorts (S6L–S6S Fig). Notably, all loci except 6q21 showed orders of magnitude higher significance in C-GWAS than in MinGWAS (Fig 1B), with rs17034666 at 2q12.3 *EDAR* being the most extreme example (from $P_{\mathrm{MinGWAS}}$ = 1.09e-11 to $P_{\mathrm{C\text{-}GWAS}}$ = 1.38e-24). The East Asian-specific missense variant of *EDAR* (*EDARV370A*, rs3827760) had been previously associated with a wide range of endoderm-derived phenotypes, such as chin protrusion [15], hair shape [16], hair thickness [17], sweat glands [18], size of feminine breasts [18], and shovel incisors characteristic [19,20]. However, this SNP was removed from our C-GWAS due to its very low MAF in European samples. The boost of significance at 2q12.3 *EDAR* is an example of the increased power of C-GWAS in detecting multi-trait effects. A multivariable fitting analysis in the RS cohort based on individual-level data showed that the lead SNPs from C-GWAS-significant loci explained on average 1.5 times and up to 3.1 times more phenotypic ear variance than the SNPs from MinGWAS-significant loci did (Fig 1H–1I).

## Integration with previous literature knowledge

The two previous single-trait GWASs on qualitative ear morphology [1,2] identified 58 autosomal SNPs at 50 loci mainly associated with earlobe features. We looked up these 58 SNPs in three of our cohorts (RS, TwinsUK, and NSPT) not considering CANDELA and TZL because these two cohorts were used in the previous GWASs. For 49 SNPs (nine SNPs were non-polymorphic in at least one dataset and thus got excluded from this analysis), both C-GWAS (*P* = 1.1e-8) and MinGWAS (*P* = 2e-5) p-values highly significantly deviated from the null, while C-GWAS p-values obviously deviated further from the null than MinGWAS p-values did (Fig 1G). Under the nominal significance level, C-GWAS re-identified a larger number of the previously established loci than MinGWAS did (16 vs. 13). In addition, all our nominally significant associations involving earlobe landmarks (S5 Table) were consistent with the findings from previous studies of earlobe features, i.e., they were previously associated with earlobe features and the most significant ear phenotype in our study involved earlobe landmarks too. These results further confirm the increased statistical power of C-GWAS compared to the traditionally used MinGWAS.

Both the ear and face belong to craniofacial phenotypes for which shared genetic effects maybe expected. Therefore, we examined the 238 distinct genetic loci previously associated with human facial shape variation [8,15,21–30] in our C-GWAS results for quantitative ear morphology. Two of these face loci, which harbor *TBX15* and *INTU*, respectively, (S6 Table)

showed study-wide significant association in our ear C-GWAS. On the other hand, we saw that a substantial proportion of the ear-associated loci identified in the present ear study (6/16) showed genome-wide significant association with facial shape variation in previous face GWASs (Fig 1F). The finding of a larger proportion of ear-associated loci implicated in facial shape than face-associated loci implicated in ear morphology is consistent with the longer span of early development of the face (4th to 9th gestation week) compared to the outer ear (6th and 7th gestation week) [31,32]. Furthermore, more than half (9/16) of our C-GWAS-identified ear loci (1p12 *TBX15*, 2q12.3 *EDAR*, 2q31.1 *SP5/MYO3B*, 3q23 *MRPS22*, 6q21 *ATG5*, 8q24.13 *HAS2*, 9q33.1 *ASTN2*, 10q22.2 *C10orf11*, 20q11.22 *UQCC1*), which were particularly associated with earlobe phenotypes in our study, were previously reported to have association with male pattern baldness in the GWAS catalog [10], and two of our ear loci, including 2q12.3 *EDAR* and 6q21 *ATG5*, were previously associated with mono eyebrow [21] (S7 Table). Male pattern baldness and mono eyebrow also reflect surface ectoderm-derived phenotypes. These results suggest that surface ectoderm-derived phenotypes such as facial shape, ear morphology, male pattern baldness, and mono eyebrow share genetic factors, which may be expected, but has not been empirically demonstrated before. These findings provide further evidence of the genetic overlap between different craniofacial phenotypes and highlights the importance of studying multiple traits together to gain a more comprehensive understanding of the genetic basis of craniofacial variation.

### Functional annotations

Several lines of evidence support the functional implications of the 16 discovered ear-associated loci, including 8 novel loci. A gene ontology enrichment analysis highlighted 6 ear development-related biological process terms (FDR < 0.01, Tables 1 and S8) including ear development, inner ear development, development and morphogenesis of the skeletal system, and embryonic organ development and morphogenesis. The majority of the lead SNPs (10/16) showed evidence of regulatory activity in the 3DSNP database [33] (S9 Table). Four out of the 16 loci showed positive enhancer activity in transgenic mice supported by the spatial pattern of expression located in ears/branchial arch/craniofacial [34]. The majority (14/16) of nearby genes or 3D interaction genes has been associated with abnormal ear/craniofacial phenotypes in various databases [35]. Most (15/16) of nearby genes or 3D interaction genes were expressed in branchial arch and embryo ectoderm in mice [36,37]. These lines of evidence strengthen the reliability of our novel loci and support that ear and cranial morphogenesis share a substantial proportion of genetic factors. These findings provide further support for the functional relevance of the identified loci and highlight the importance of studying multiple traits together to gain a more comprehensive understanding of the genetic basis of craniofacial development.

Embryonic cranial neural crest cells (CNCCs) are temporary, migratory stem cells that play a crucial role in the formation of the ear during embryonic development. We compared the expression of 16 genes located near the 16 ear-associated SNPs with a random set of 16 genes selected near randomly chosen and frequency-matched 16 SNPs in CNCCs and 49 other cell types for 10000 replicates. Compared with the randomly selected gene sets, this set of 16 targeted genes near the ear-associated SNPs had significantly higher expression in 23 types of cells ($P < 0.05$, Fig 2A), including those involved in the formation of constituent of ear, such as CNCCs, articular chondrocyte, osteoblast, skin fibroblast, and adipose mesenchymal stem cell. In addition, we performed a heritability enrichment analysis using the S-LDSC method [38] with respect to active regulatory regions based on our C-GWAS results of Europeans (RS and TwinsUK). This analysis showed that the S-LDSC coefficient Z-scores in mesenchymal stem cells, osteoblast primary cells, adult dermal fibroblast primary cells, adipose derived

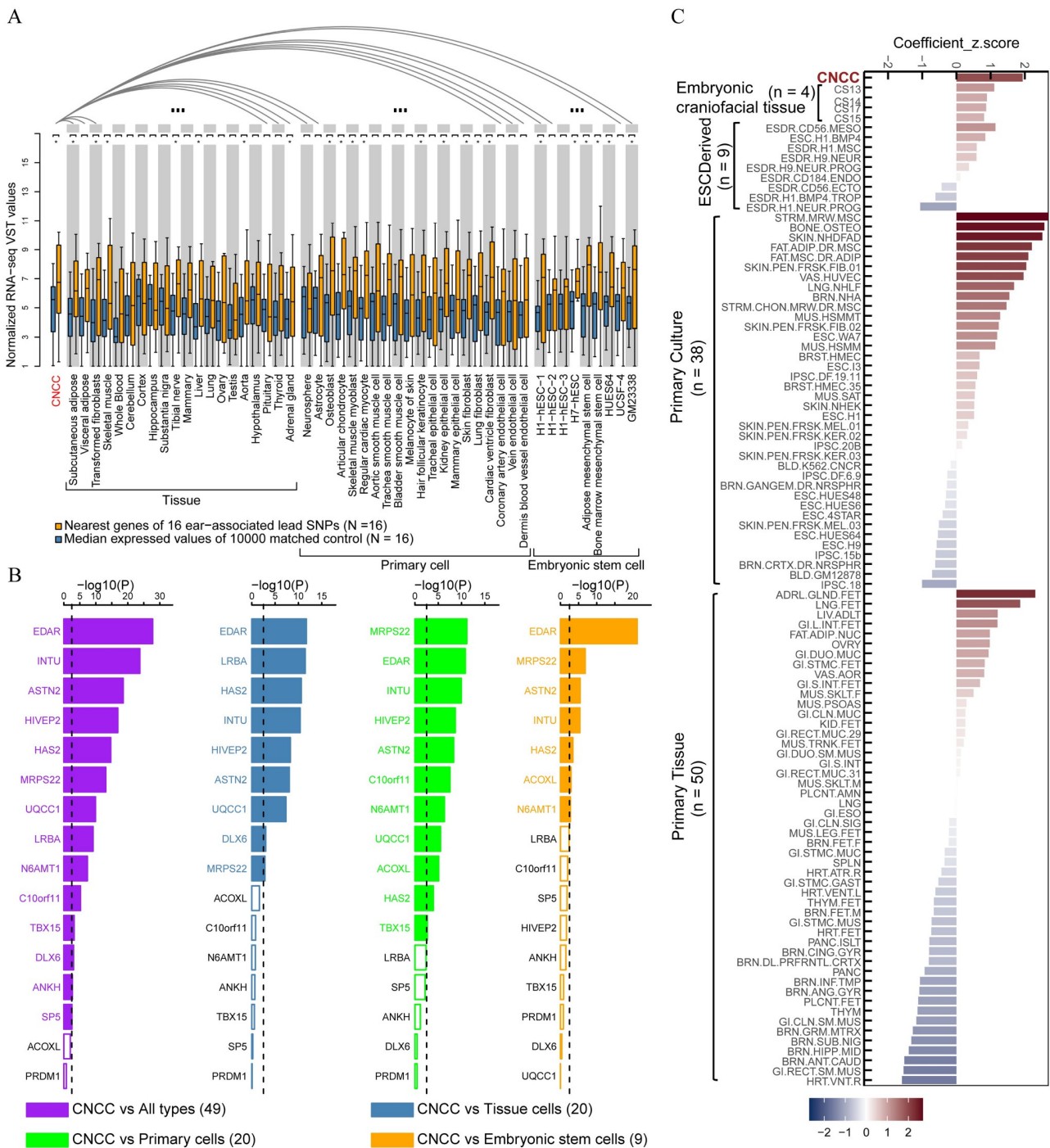

**Fig 2. Differential expressions of 16 genes near 16 ear-associated loci in 50 cell types.** (**A**) Boxplots of normalized RNA-seq VST values for the 16 genes (in orange, details of genes shown in lower B plot) and 16 median gene expression value as control (randomly matched 1e-4 times using SNPsnap, in blue). Expression differences between the control genes and the 16 genes were tested in each cell type using the unpaired Wilcoxon rank-sum test. The expression of the 16 genes in CNCCs was also iteratively compared with that in all other cell types using paired Wilcoxon rank-sum test. Statistical significance was indicated: *P < 0.05. (**B**) Normalized RNA-seq VST values in CNCCs were compared with those in other 49 types of cells (purple), in 20 tissue cell types (blue), in 20 primary cell types (green), and in nine embryonic stem cell types (orange), using one-sample Student's t-test. Dotted line represents Bonferroni corrected significant threshold (*P < 2.27e-3*). Significant gene labels are depicted in color, non-significant gene labels in black. (**C**) Partitioned heritability enrichments based on cell-type-specific regulatory annotations (More details see in Methods). Heritability enrichment Z-scores, as estimated by stratified linkage disequilibrium score regression (S-LDSC) of the C-GWAS summary data for GWASs of 136 ear traits. Trait abbreviations as in S10 Table.

mesenchymal stem cell cultured cells, mesenchymal stem cell derived chondrocyte cultured cells, all ranked within the top 5% in a total of 102 types of cells and CNCCs slightly behind them (ranked 9th) (Fig 2C). These cell types from the heritability enrichment analysis are consistent with those where the 16 genes showed preferential expression. These cell types play important roles in ear morphogenesis, supporting the reliability of our findings. In addition, 14 out of these 16 genes showed preferential expression in CNCCs compared to 49 other cell types (Fig 2B). The genes near the eight newly discovered loci showed preferential expression in CNCC, highly consistent with the preferential expression pattern of the genes near the 8 previously established ear-associated loci. Interestingly, several of these genes (*EDAR*, *INTU*, and *HAS2*) have previously been linked to facial morphology [8], supporting the idea that genetic effects shaping the ear and face originate during early embryogenesis. These findings provide a priority list for future in-vivo studies on genes involved in ear variation and embryo surface ectoderm-derived phenotypes.

### Ear effects in *Intu* and *Tbx15* mutant mice

Here, we examined adult ear morphology in *Intu* and *Tbx15* mouse mutants using one-step CRISPR/cas9 mediated gene editing experiments to assess the functional significance of *Intu* and *Tbx15* expression during ear development. Our breeding experiments generated 18 F2 9-weeks sexually mature *Intu* mice i.e., 10 heterozygous *Intu*$^{+/-}$, 8 wild-type *WT* (*Intu*$^{+/+}$), while homozygous loss-of-function of *Intu* was lethal. Quantitative assessment of mouse ear shape (assessed by PC analysis of 21 three-dimensional landmark coordinates) revealed significant differences (linear regression $P_{PC3}$ = 1.5e-3) (Fig 3C) between the heterozygous *Intu* littermates and *WT* mice. The *Intu* genotype significantly associated with D6_14 ($P$ = 2.1e-3, Beta = 1.78) and D3_6 ($P$ = 3.3e-3, Beta = 1.74) (Fig 3E and S11 Table) after FDR correction. Notably, in humans, the top *INTU* variant rs57788627 was nominally significantly associated with ear phenotypes L3-L6 and L3-L7, which respectively corresponds to D6_14 and D3_6 in mice ($P$ = 1.9e-5, Beta = 0.07; $P$ = 8.7e-5, Beta = 0.06) (Fig 3E and S11 Table). Overall, the ears of the heterozygous *Intu* mutant mice were shorter than those of the *WT* mice consistently show in the result of PC3 and distance phenotypes (Fig 3D–3F). Furthermore, the heterozygous *Intu* mutant mice showed a significant trend of reduction in body length and fore and hind limb length (S8 Fig), which was consistent with previous findings [13].

Our breeding experiment generated 37 F2 9-weeks adult *Tbx15* mice i.e., 9 homozygous *Tbx15*$^{-/-}$, 18 heterozygous *Tbx15*$^{+/-}$, 10 wild-type *WT* (*Tbx15*$^{+/+}$). Previous gene editing experiments have already shown that *Tbx15*$^{-/-}$ mutant mice demonstrate facial variation, loss of weight, shorter limbs and a distinct "droopy ear" feature [39]. In our study, besides the facial differences reported previously, ear differences were obvious. In addition to the distinct "droopy ear" feature, the *Tbx15* genotype was significantly associated with the ear landmarks PC1 ($P$ = 3.2e-3) and PC4 ($P$ = 1.9e-3) (S9C Fig). Overall, the ears of the *Tbx15* mutant mice were obviously longer and wider than those of the *WT* mice (S9E and S9F Fig).

## Discussion

In this study of the genetic architecture of ear morphology, we developed a CNN-based deep learning model to automatically locate 17 anatomical ear landmarks, allowing us to quantify 136 ear traits in 14,921 individuals from three continents. We then used a recently developed C-GWAS method [4] to combine GWASs of ear traits. These efforts allowed us to identify 16 genetic loci associated with multiple ear traits, including 8 novel and 8 previously reported loci [1,2]. Bioinformatic analysis supports the functional role of the newly identified genes in ear morphogenesis, and gene editing experiments in mice showed that a gene desert near *INTU*

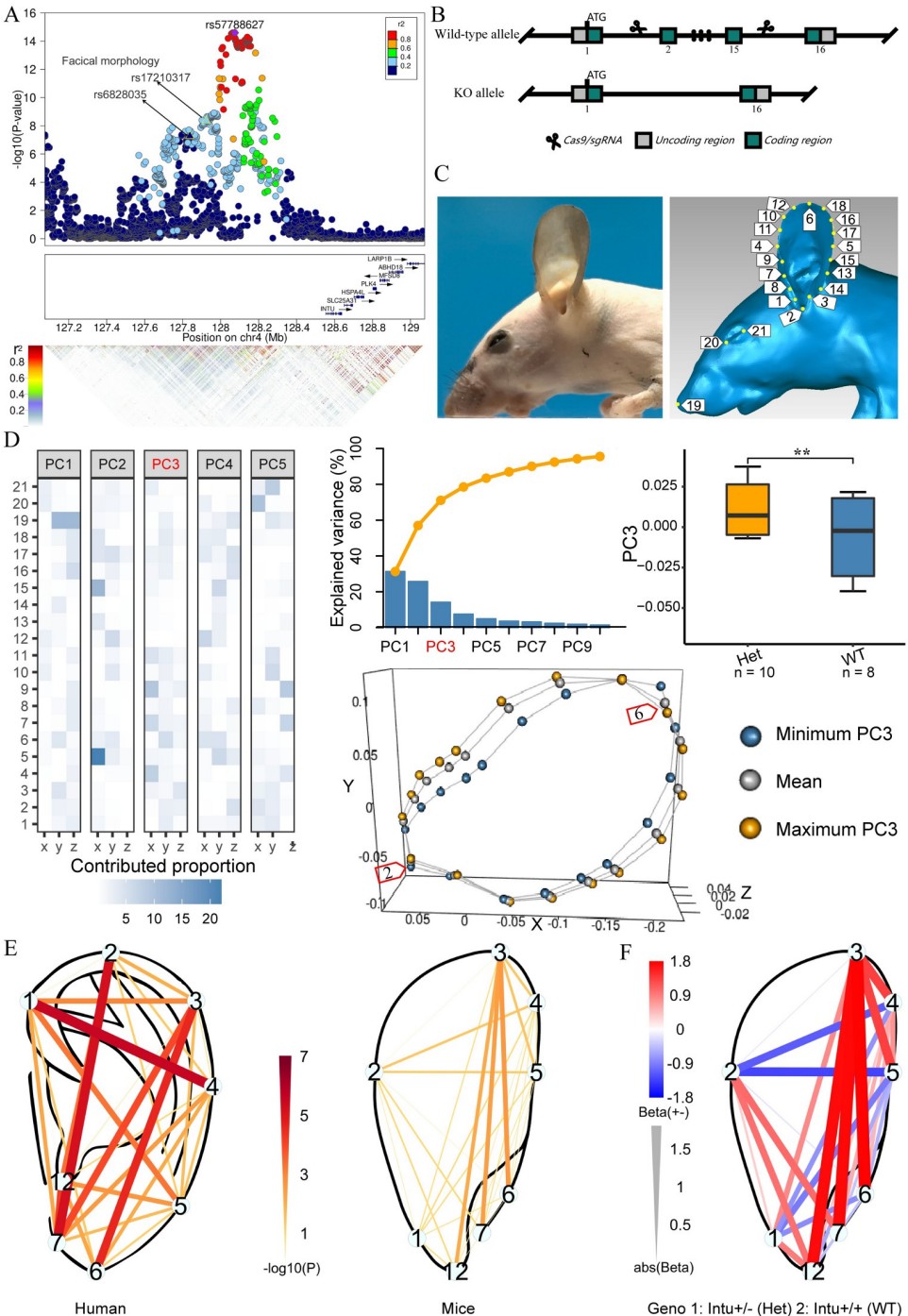

**Fig 3. In-vivo mouse models of *INTU* deficiency.** *Intu* mutant mice (*Intu*$^{-+/-}$, n = 10, 9 weeks) vs. C57BL/6 wild type control mice (*WT*$^{+/+}$, n = 8, 9 weeks) were compared for ear and body morphological differences (for details of *Tbx15* mice see S9 Fig). (**A**) Regional association plot for the associated locus nearby *INTU*, and the other two SNPs at this locus, which have been recently reported the association with face morphology, were also marked. (**B**) The schematic diagram of the one-step CRISPR/Cas9 technology used in *Intu* knockout mice (see S10 Fig for details). (**C**) Example of left profile craniofacial photo of *Intu*$^{+/-}$ mutant mice with removed hair (left); and 21 craniofacial landmarks pattern in 3D mice image (right). (**D**) The principal component analysis for 21 landmarks of *Intu*$^{+/-}$ mutant mice and *WT*$^{+/+}$ mice. The left layer shown the detailed contributed proportion of 21 landmarks to the first 5 principal components. The upper left shown the cumulative variance is explained by the first 10 PCs. The significant association between the genotype and PC3 shown in upper right (* $P < 0.05$, ** $P < 0.01$, *** $P < 0.001$). The bottom shown the maximum PC3-, minimum PC3- and mean ear shapes (the labels on the mouse ear were consistent with the FigC). (**E**) The

pattern of genetic association in humans (left) and in mice (right). (**F**) Effect of *Intu* knock-out on ear phenotypes in mice (blue for effect of heterozygote mutant and red for wild type). The labels on three ears were consistent with Fig S1 (more details see in Methods).

has a functional impact on ear morphology. Our findings also highlight shared genetic contributors between some surface ectoderm-derived appearance phenotypes, with an emphasis on ear and facial morphology, male pattern baldness, and mono eyebrow.

The novel ear-associations at 5 of the 8 loci we identified here i.e., 4q28.1 *INTU*, 5p15.1 *ANKH*, 8q24.13 *HAS2/HAS2-AS1*, 10q22.2 *C10orf11*, and 20q11.22 *UQCC1/GDF5* are supported by other evidence, whereas for 3 novel ear genes (2q13 *ACOXL*, 9q33.1 *ASTN2*, and 21q21.3 *N6AMT1*) no such evidence is currently available. The strong evidence for *INTU* and *HAS2* has been detailed above. The gene *ANKH*, which regulates inorganic pyrophosphate transport, is located near the lead SNP rs10062331 at 5p15.1. *ANKH* has been linked to diseases such as Craniometaphyseal Dysplasia (OMIM: 123000) and Chondrocalcinosis 2 (OMIM: 118600), both of which are characterized by abnormal development of bones and connective tissue. *ANKH*'s association with skeletal disorders suggests it may play a role in ear cartilage development. The lead SNP rs10824309 at 10q22.2 is an intronic variant of *C10orf11*, also known as *LRMDA*. This gene has been associated with craniofacial traits such as adolescent idiopathic scoliosis [40], heel bone mineral density [41], and chin shape [21]. This provides additional support for the novel association between 10q22.2 and ear development as craniofacial and ear development both stem from a common origin during early development. The lead SNP rs2378353 located at 20q11.22 is within the intron of *UQCC1*, which encodes a transmembrane protein involved in the assembly of the ubiquinol-cytochrome c reductase complex. Polymorphisms in this gene have been linked to human height [42] and osteoarthritis [43], and rs2378353 has been found to have a significant impact on the splicing of *UQCC1* in muscle-skeletal tissue, as well as its expression in multiple tissues including fibroblasts, muscle-skeletal tissue, and skin [44]. *UQCC1* is expressed in the branchial arches and head surface ectoderm during early embryonic development in mice [45]. These findings, combined with the differential expression of *UQCC1* in cranial neural crest cells, suggest that rs2378353 may play a role in ear morphogenesis by regulating the expression of *UQCC1* during early development. The functional information for the other 3 novel ear-associated loci identified in this study is limited.

Eight of the 16 identified loci we identified here with association with quantitative ear traits have been previously reported to be associated with qualitative ear features (mainly earlobe features) in single-trait GWASs, including 1p12 *TBX15* [2], 2q12.3 *EDAR* [1,2], 2q31.1 *SP5/MYO3B* [1,2], 3q23 *PISRT1/MRPS22* [1,2], 4q31.3 *LRBA* [1,2], 6q21 *PRDM1/ATG5* [2], 6q24.2 *LOC153910/HIVEP2* [2], and 7q21.3 *DLX6/SEM1* [2]. Two of these (4q31.3 *LRBA* and 7q21.3 *DLX6/SEM1*), which were solely identified by C-GWAS in our study, were previously identified in a single GWAS of earlobe attachment involving ~70,000 samples; hence, in a study that was five times larger than our current study. That besides the 5x smaller sample size, our study identified 8 novel loci, half of which were identified only by C-GWAS, confirms the power gain of C-GWAS in detecting small but multi-trait effects, as previously highlighted also for human face shape traits [4].

A significant portion of the 16 ear-related loci discussed here have been shown to have strong ties to human facial structure [8,15,21,22,30]. This indicates that both ear and face morphologies are influenced by common genetic factors, likely stemming from shared developmental processes in the craniofacial area. Additionally, a considerable number of the 16 ear-related loci were also previously linked to male pattern baldness, and two loci were linked to

mono-eyebrow [21]. These findings suggest that not only do ears and face share genetic factors, but ears, baldness, and mono-eyebrow also share genetic factors, and it is possible that other craniofacial traits may also be influenced by similar genetic factors. Further investigation is needed to fully understand this.

Our study sheds light on the genetic basis of the Darwin tubercle, a special ear phenotype that is equivalent to the ear tip of other mammals and represents a trace of human evolution [11,46]. Previous genetic studies on this trait have been limited and have not produced significant results. Our findings indicate that the *HAS2* gene influences the L3-L15 phenotype, which roughly represents the Darwin tubercle, marking the beginning of understanding this intriguing aspect of human evolution.

This study involved cohorts belonging to three continental populations from Europe, Asia, and America. For detecting cross-population allelic effects, C-GWAS demonstrated noticeably improved power than MinGWAS as evident by boosting the significance of known ear-associated loci while keeping the same study-wide type-I error rate as MinGWAS. However, because our approach required the a priori exclusion of SNPs not polymorphic in one of the studied cohorts, population-specific association signals, could not be detected. Another flip-side caveat of our study, in which we maximized the sample size used for discovery, is that no additional population samples were available for direct replication of our novel findings. However, that half of the loci we identified represent previously known ear loci that we re-discovered with our approach, puts confidence in our approach and thus also in the true-positive status of the newly identified loci. Additional confidence is provided by other evidence we accumulated from the literature for most of the novel loci as well as by direct functional evidence in mice for one novel gene (*INTU*) and one previously reported gene (*TBX15*). Nevertheless, future studies in independent population samples from different continental populations are warranted to further verify our novel findings.

Our study represents a landmark in the field of genetics as it is the first to examine the genetic basis of quantitative ear morphology using a deep-learning-based CNN phenotyping approach. By combining the results of 136 single ear trait GWASs conducted in well-sized population samples of different ancestral origins, our C-GWAS method enabled us to identify eight new loci and confirm eight previously reported loci that play a role in ear morphology. These loci impact a wide range of ear phenotypes, demonstrating the increased power of C-GWAS in detecting multi-trait effects. Furthermore, our findings suggest that many facial and ear traits share a substantial proportion of genetic determinants, derived from the surface ectoderm. Our study significantly advances the understanding of the genetics of human ear morphology and provides a list of promising candidate genes for further functional studies.

## Materials and methods

### Ethics statement

This study includes five cohorts, all cohorts were approved by the Ethics Committee and all participants provided written informed consent to participate in the study. Details in following: The Rotterdam Study (RS) has been approved by the Medical Ethics Committee of the Erasmus MC (registration number MEC 02.1015) and by the Dutch Ministry of Health, Welfare and Sport (Population Screening Act WBO, license number 1071272-159521-PG). The TwinsUK Study (TwinsUK) has been approved by the St. Thomas' Hospital Local Research Ethics Committee. The Taizhou Longitudinal Study (TZL) has been approved by the Ethics Committee of Human Genetic Resources at the Shanghai institute of life Sciences, Chinese Academy of Sciences (ER-SIBS-261410). The National Survey of Physical Traits Study (NSPT) has been approved by the Ethics Committee of Human Genetic Resources of School of Life

Sciences, Fudan University, Shanghai (14117). The Consortium for the Analysis of the Diversity and Evolution of Latin America (CANDELA) Study has been approved by ethical committee at universities in all samples countries: the Universidad Nacional Autonoma de Mexico (Mexico), the Universidad de Antioquia (Colombia), the Universidad Peruana Cayetano Heredia (Peru), the Universidad de Tarapaca (Chile), the Universidade Federal do Rio Grande do Sul (Brazil) as well as at the University College London (UK).

## Study cohorts

This study used a total of 14,921 samples that passed quality control from five cohorts of multi-ethnic ancestries from Europe (the Rotterdam Study, RS, $N$ = 3,675; the TwinsUK study, $N$ = 1,065), Asia (the Taizhou Longitudinal Study, TZL, $N$ = 2,348; the National Survey of Physical Traits study, NSPT, $N$ = 2,487), and Latin America (the CANDELA study, $N$ = 5,346). Details about these five cohorts are included below.

## The Rotterdam Study

The Rotterdam Study (RS) is a population-based prospective study of individuals aged $\geq$ 45 years living in a suburb of Rotterdam, the Netherlands. Details regarding the cohort profile have been described elsewhere [47]. The Rotterdam Study has been approved by the Medical Ethics Committee of the Erasmus MC (registration number MEC 02.1015) and by the Dutch Ministry of Health, Welfare and Sport (Population Screening Act WBO, license number 1071272-159521-PG). The Rotterdam Study has been entered into the Netherlands National Trial Register (NTR; www.trialregister.nl) and into the WHO International Clinical Trials Registry Platform (ICTRP; www.who.int/ictrp/network/primary/en/) under shared catalogue number NTR6831. All participants provided written informed consent to participate in the study. A total of 5,604 participants not wearing make-up, cream nor jewelry, were photographed using a Premier 3dMD face3-plus UHD camera (3dMD, Atlanta, Georgia, USA). Each person's 3D avatar is rendered from three 2D high resolution photos taken from predefined angles, i.e., from front-top, left-down, and right-down directions. Ears in the left-down and right-down photos were manually segmented and chopped for subsequent phenotyping. Genotyping was carried out using the Infinium II HumanHap 550K Genotyping BeadChip version 3 (Illumina, San Diego, California USA). All SNPs were imputed using MACH software (www.sph.umich.edu/csg/abecasis/MaCH/) based on the 1000-Genomes Project reference population information [48]. Genotype and individual quality controls have been described in detail previously [49]. In current study, after all phenotype and genotype quality controls, this study included 3,675 individuals of RS.

## TwinsUK Study

The TwinsUK Study (TwinsUK) included 1,153 phenotyped participants (all female and all of European ancestry) within the TwinsUK adult twin registry based at St. Thomas' Hospital in London. Volunteers gave written informed consent under a protocol reviewed by the St. Thomas' Hospital Local Research Ethics Committee. Participants were photographed using a Premier 3dMD face3-plus UHD camera (3dMD, Atlanta, Georgia, USA), and 3D avatars were rendered from two 2D photos taken from the left and right directions. Ears in the 2D photos were manually segmented and chopped for subsequent phenotyping. Genotyping of the TwinsUK cohort was done with a combination of Illumina HumanHap300 and HumanHap610Q chips. Intensity data for each of the arrays were pooled separately and genotypes were called with the Illuminus32 calling algorithm, thresholding on a maximum posterior probability of 0.95 as previously described [50]. Imputation was performed using the IMPUTE

2.0 software package using haplotype information from the 1000-Genomes Project (Phase 1, integrated variant set across 1,092 individuals, v2, March 2012). After all phenotype and genotype quality controls, the current study included a total of 9.35 million autosomal SNPs (MAF > 0.01, imputation R2 > 0.3, SNP call rate > 0.97, HWE > 1e-6) and 1,065 individuals of TwinsUK.

## Taizhou Longitudinal Study

The Taizhou Longitudinal Study (TZL) includes Han Chinese sampled in Taizhou, Jiangsu province in 2014 [51]. In total, 2,600 individuals were enrolled. The TZL was approved by the Ethics Committee of Human Genetic Resources at the Shanghai institute of life Sciences, Chinese Academy of Sciences (ER-SIBS-261410). All participants had provided written consent. Participants were photographed using a Canon EOS700D camera. Four digital photographs of the face: left side (−90˚), frontal (0˚)*2, right side (90˚) were taken from ∼ 1.5 meters. Ears were cropped from the left side facial photographs. All samples were genotyped using the Illumina HumanOmniZhongHua-8 chips, which interrogates 894,517 SNPs. Individuals with more than 5% missing data, related individuals, and the ones that failed the X-chromosome sex concordance check or had ethnic information incompatible with their genetic information were excluded. SNPs with more than 2% missing data, with a minor allele frequency smaller than 1%, and the ones that failed the Hardy–Weinberg deviation test ($P <$ 1e-5) were also excluded. After applying these filters, we obtained a dataset of 2,600 samples with 776,213 SNPs. The chip genotype data were firstly phased using SHAPEIT [52]. IMPUTE2 [53] was then used to impute genotypes at non-genotyped SNPs using the 1000 Genomes Phase 3 data as the reference panel. After all phenotype and genotype quality controls, the current study included a total of 7,057,720 imputed and genotyped SNPs and 2,348 individuals of TZL.

## The National Survey of Physical Traits Study

The National Survey of Physical Traits Study (NSPT) is part of the National Science & Technology Basic Research Project, which contained four sub-cohorts collected from three different regions of China in different years: Taizhou, Jiangsu province in 2015 and 2019, Zhengzhou, Henan province in 2017, Nanning, Guangxi province in 2018. The NSPT project was approved by the Ethics Committee of Human Genetic Resources of School of Life Sciences, Fudan University, Shanghai (14117). All participants provided written informed consent. Participants were photographed using a Canon EOS700D camera. Ten digital photographs of the face: left side (−90˚)*2, left angle (−45˚)*2, frontal (0˚)*2, right angle (45˚)*2, right side (90˚)*2 were taken from ∼ 1.5 meters. Photos in each direction include one with eyes open and one with eyes closed. Ears were cropped from the left side facial photographs with open eyes. DNA was extracted from blood samples using the MagPure Blood DNA KF Kit. The DNA samples were genotyped on the Illumina Infinium Global Screening Array that investigates 707,180 variants which is a fully custom array designed by WeGene (https://www.wegene.com/). Individuals with more than 5% missing data, related individuals, and the ones that failed the X-chromosome sex concordance check or had ethnic information incompatible with their genetic information were excluded. In total, 2,487 individuals were enrolled (15HanTZ: $N =$ 404; 17HanZZ: $N =$ 644; 18HanNN: $N =$ 1,119; 19HanTZ: $N =$ 320). The genotype data were phased using SHAPEIT [52] and imputed to the 1000 Genomes Project Phase 3 reference panel using IMPUTE2 [53]. Variants exclusion criteria included INFO < 0.8, certainty score < 0.9, MAF < 0.02, SNP-wise call rate > 5%, and deviation from Hardy-Weinberg equilibrium ($P < 1 \times 10^{-6}$). After all phenotype and genotype quality controls, the current study included a total of 8,018,212 imputed and genotyped SNPs and 2,487 individuals of NSPT.

## CANDELA Study

The Consortium for the Analysis of the Diversity and Evolution of *Latin America* (CANDELA) Study [54] consists of 6,630 volunteers recruited in five Latin American countries (Brazil, Colombia, Chile, Mexico and Peru). Ethics approvals were obtained from ethical committee at universities in all samples countries: the Universidad Nacional Autonoma de Mexico (Mexico), the Universidad de Antioquia (Colombia), the Universidad Peruana Cayetano Heredia (Peru), the Universidad de Tarapaca (Chile), the Universidade Federal do Rio Grande do Sul (Brazil) as well as at the University College London (UK). All participants provided written informed consent. Five digital photographs of the face: left side ($-90°$), left angle ($-45°$), frontal ($0°$), right angle ($45°$), right side ($90°$) were taken from ~1.5 meters at eye level using a Nikon D90 camera fitted with a Nikkor 50 mm fixed focal length lens. Ears were cropped from the left side facial photographs. DNA samples were genotyped on Illumina's Omni Express BeadChip. After applying quality control filters 669, 462 SNPs and 6,357 individuals were retained for further analyses (2,922 males, 3,435 females). Average admixture proportions for this sample were estimated as: 48% European, 46% Native American and 6% African, but with substantial inter-individual variation. After all genomic and phenotypic quality controls this study included 6,238 individuals. The genetic PCs were obtained from the LD-pruned dataset of 93,328 SNPs. These PCs were selected by inspecting the proportion of variance explained and checking scatter and scree plots. The final imputed dataset used in the GWAS analyses included genotypes for 9,143,600 SNPs using the 1000-Genomes Phase I reference panel. After all phenotype and genotype quality controls, the current study included a total of 9,143,600 imputed and genotyped SNPs and 5,346 individuals of CANDELA.

## Deep learning based ear phenotyping

Our computer pipeline for quantitative ear phenotyping, which includes automated ear detection, segmentation, landmarking, and phenotype acquisition is available at https://github.com/Fun-Gene/EarPhenotyping. In all cohorts, we used the same pipeline to minimize potential variation caused by different methodology. Specifically, we used the previously established two-stage ear landmark detector consisting of two CNNs deep-learnt from ~15,000 labeled ear images [3]. Both CNNs had the same architecture consisting of alternating 3 convolutional layers and 3 max-pooling layers followed by 3 fully connected layers. The 1st CNN detects various orientations and sizes and rectifies all coordinates at a coarse-grained level. The 2nd CNN was trained in a more controlled scenario for accurate landmarking of 55 ear landmarks. The performance of this two-stage landmark detector has been verified in various scenarios of difficulty that represents the state-of-the-art in ear landmark detection. In our application, we double checked all resultant landmarks from all images by eye-balling and removed those low-quality images that failed in CNN-based landmarking. After obtaining the coordinates of 55 landmarks, Generalized Procrustes Analysis (GPA) was used to remove affine variations due to shifting, rotation, and scaling. We then focused on the 17 most anatomical meaningful landmarks covering the entire ear (S1A Fig and S12 Table) and from those, calculated 136 inter-landmark distances as ear phenotypes in the genetic studies. Outliers greater than 3 standard deviations were removed and Z-transformed values were used in all subsequent analyses.

For quality control purposes, a trained rater carefully labeled the 17 landmarks on both the left and the right ear in 50 randomly selected and shuffled images. Pearson's correlations were calculated between the left and the right ear phenotypes from this rater, which were compared with the left-right correlations from the CNN approach. Within the same ear (the left side only), Pearson's correlations between the rater and the CNN approach were calculated.

### Genetic correlation, heritability, GWASs, meta-analyses

Genetic correlations were estimated from all SNPs using GEMMA [55] based on a multivariate linear mixed model. Unsupervised hierarchical clustering analysis is conducted using 1-abs (correlation) as a dissimilarity matrix, and each iteration is updated using the Lance-Williams formula [56]. Twin heritability was estimated using phenotype correlations in monozygotic (MZ) and dizygotic twins (DZ), $h^2 = 2(r(MZ)-r(DZ))$. SNP-based heritability was estimated from all GWAS SNPs using the restricted maximum likelihood estimation in GCTA [57].

GWASs were independently carried out in each cohort. GWASs in unrelated individuals (RS, TZL, NSPT, and CANDELA) were performed using linear models assuming an additive allele effect adjusted for covariates sex, age, BMI, and top 5–10 genomic PCs using PLINK 1.9 [58]. GWASs in TwinsUK (females only) were performed using GEMMA [55], which implements an LME model with an empirical genetic relatedness matrix to account for cryptic pedigree and population structure. All cohorts were aligned according to the genome-build GRCh37.p13. Meta-analyses of all cohorts were conducted using the inverse variance fixed-effect model in PLINK 1.9. All statistical analyses were conducted using the R Environment for Statistical Computing (version 3.5.2) unless otherwise specified.

### C-GWAS

In the application of C-GWAS analysis on 136 ear meta-analyses, we focused on SNPs with an observable frequency (MAF > 0.01) in three different continental groups (European, East Asian, and Latin American). The C-GWAS is an R library that is freely available at https://github.com/Fun-Gene/CGWAS. In the current application, default parameters were used. The summary statistics of the 136 meta-analyses were used as the input of C-GWAS. Details of C-GWAS method has been described previously [4]. In brief, the null hypothesis (H0) for a SNP under testing is the absence of any allelic effect on all traits, and the alternative hypothesis (H1) is that its allelic effects deviate from 0 for at least one of the multiple traits. C-GWAS incorporate two different tests originated from either the effect based inversed covariance weighting or the truncated Wald test to maximize statistical power. All resultant P-values from C-GWAS are adjusted using the getCoef function implemented in C-GWAS, which performs simulations (n simulations = 1e8) to guarantee that the null distribution of C-GWAS follows the uniform distribution in all quantiles. This simulation analysis was also applied for adjusting for minimal p-values of meta-analysis of 136 traits, and the adjusted minimal P-values are abbreviated as MinGWAS. Therefore, C-GWAS and from MinGWAS are directly comparable with each other and with any standard single-trait GWAS, so that the traditional genome-wide significance threshold of 5e-8 corresponds to our study-wide significance threshold. C-GWAS completed the analysis of 136 traits and 4,803,785 SNPs within 2 hours with 16 threads in parallel and peak memory usage of 32 GB.

### Post-GWAS analyses

SNPs and nearby genes were annotated using ANNOVAR [59]. Enrichment analysis of biological processes, molecular functions, and cellular components were conducted using Metascape [60] based on the Gene Ontology (GO) database. Regulatory activities of associated SNPs and 3D interacting genes were explored using the 3DSNP database [33]. Enhancer activities and embryonic expression patterns at the associated loci were examined in transgenic mice in the Vista Enhancer Browser database [34]. Potential functional links between nearby genes, 3D interacting genes and ear/craniofacial features were examined in the Harmonizome database [35] and gene expression patterns in the branchial arch and embryonic ectoderm were examined in the MGI database [36,37]. Gene expressions in cranial neural crest cells (CNCCs) were

compared between genes of interest and background genes over the genome using RNA-seq data from Prescott et al. [61], GTEx [44], and ENCODE [62]. We attained all annotated nearest genes of the 1e-4 sets of matched SNPs for 16 lead SNPs using SNPsnap [63], the 16 genes with the median expression value separately in 50 cell types were as the correspondent control genes. Partitioned heritability enrichments based on CNCCs [61], 4-stages of embryonic craniofacial tissues [64] and 97 cell types form Roadmap Epigenomics resource [65,66]. The annotations for CNCCs, embryonic craniofacial tissues and other cell types refer the previous study [67] using S-LDSC [38]. For CNCCs, we downloaded and processed H3K27ac and ATAC data, and all data from different replicates. Peaks were called using MACS2 [68]. We iteratively obtained and combined the most reliable peaks based on a 50% peak overlap rate for all replicates from of H3K27ac and ATAC data using BEDtools. [69]. Based on the combined peaks, we used ROSE [70] to infer enhancers including super-enhancers and annotated all enhancer regions (S14 Table). For embryonic craniofacial tissues, we combined all regions with the following annotations from the 25-state chromHMM model: 'Enh', 'TxReg', 'PromD1', 'PromD2', 'PromU' and 'TssA'. For other cell types, we combined all regions with the following annotations from the 15-state chromHMM model: '1_TssA', '2_TssAFlnk', '7_Enh' and '6_EnhG'. Each annotation was individually added to the baseline LD model [38]. Also, we use the C-GWAS summary data for 136 meta-analysis in European populations including RS and TwinsUK. Pleiotropic associations were looked up in GWAScatalog based on the region (basepair position of lead SNP +- 500kb) and annotated gene [71].

## CRISPR-Cas9-mediated gene editing in mouse

We generated C57BL/6J *Intu* knockout mice using the one-step CRISPR/Cas9 method [72] and tightly followed the steps described in a previous study of *Tbx15* and *Pax1* genes [39]. In brief, fertilized eggs obtained from super-ovulated females (four weeks) mated with males (seven-eight weeks) were microinjected with mixtures of Cas9 mRNA and sgRNA (S10 Fig). The injected eggs were cultured to day two and transferred to female mice. We obtained only positive heterozygote mutants because homozygote mutants of *Intu* were fatal. Compared with the intact allele (402kb), the mutant allele had a reduced sequence (251kb) with a removal of exons from 2 to 15 (S10 Fig). Mice were raised in a pathogen-free environment and bred according to SPF animal breeding standards. After eight months breeding experiments, we obtained a total of 19 F2 sexually mature mice (9-weeks), including 8 wild-type and 11 heterozygote mutant mice. For *Tbx15*, we used previously generated mice [39], including 10 wild-type, 18 heterozygote, and 10 homozygote mutant mice. The use of laboratory animals (SYXK 2019–0022) was licensed by the Beijing Municipal Science and Technology Commission.

## Mouse pinna phenotyping

F2 sexually mature mice were sacrificed by cervical dislocation. Hair was removed with a razor and Weiting depilatory cream. A HandySCAN BLACK scanner was used to obtain 3D models (resolution 0.3 mm, accuracy 0.03 mm). The coordinates of 9 anatomical landmarks (3 facial and 6 ear landmarks) and 12 pseudo-landmarks were obtained using Geomagic Wrap (Fig 2C and S13 Table). The 12 pseudo-landmarks were equally spaced from the four partial curves of the mouse auricle contour (including L1-L4, L4-L6, L6-L5, and L5-L3) for better covering the contour of the auricle. The R package 'geomorph' were used for series of analysis (https://github.com/geomorphR/geomorph). GPA was used to remove the effects of translation, rotation and scaling. After superimposition of the GPA-adjusted coordinates, only the shape component remained in the aligned specimens. Mice with the outlier in PC1 were excluded then the GPA and PCA were redone. The first 10 PCs were used to represent dominant but

different dimensional variations of all landmarks. The ear shape visualization of shape variation based on the maximum and minimum PC compare to mean ear shape. The inter-landmark also distances also were visualized (S13 Table). Z-transformed variables were used for association analysis. The effect of per mutant allele on ear shape was tested using linear regression with covariate sex, weight and body length. Multiple testing was corrected using FDR method.

## Supporting information

**S1 Fig. Study design.** (A) The location of selected 17 ear landmarks. (B) Design of the current study.
(TIF)

**S2 Fig. Pearson's correlation coefficients for 136 ear phenotypes derived from different methods (auto pipeline and manual-landmarking) and different ears (left and right ears).** (A) correlation between left ear phenotypes by manual-landmarking (expert 1 vs. expert 2). (B) correlation between right ear phenotypes by manual-landmarking and auto-landmarking (expert 1 vs. auto). (C) correlation between phenotypes from left and right ears by manual-landmarking. (D) correlation between phenotypes from left and right ears by auto-landmarking.
(TIF)

**S3 Fig. Effects of sex (left) and age (right) on 136 ear phenotypes in RS.** Please note the different figure legends in these two figures.
(TIF)

**S4 Fig. Two distinct clusters of 136 ear phenotypes derived from unsupervised hierarchical clustering.** (A) Two clusters for 136 phenotypes. (B) Phenotypic (right up) and genetic correlation matrix (left down) within and between the cluster.
(TIF)

**S5 Fig. Heritability of 136 ear phenotypes estimated in TwinsUK.** (A) Twin heritability. (B) SNP-based heritability.
(TIF)

**S6 Fig. 16 ear-associated loci we identified in our current MinGWAS and C-GWAS.** The first 8 (A-H) were novel loci, others 8 were previously reported loci (I-P). Each figure includes three figures, LocusZoom (up) shows regional association plots for the top-associated ear phenotype (p values in CGWAS, except for the 6q21 PRDM1/ATG5, which solely identified by meta-analysis) with candidate genes aligned below according to the chromosomal positions (GRCh37.p13) followed by the linkage disequilibrium (LD) patterns (r2) of European. Ear map (left lower) shows the association (p values in Meta-analysis) between all ear phenotypes ($P < 1e-3$) and top-SNP identified in our analysis. Effect plot (right lower) shows effect sizes for the effect allele of top-SNPs from the association with top-associated ear phenotype in all 5 GWASs and meta-analysis.
(DOCX)

**S7 Fig. Various degrees of Darwin's tubercle.**
(TIF)

**S8 Fig. Comparison of 4 phenotypes of *Intu*[+/-] mutant mice and *WT*[+/+] mice including weight, body length, forelimb length and posterior limb length (\* $P < 0.05$, \*\* $P < 0.01$,**

**\*\*\* $P < 0.001$).**
(TIF)

**S9 Fig. In-vivo mouse models of *TBX15* deficiency.** Homozygous $Tbx15^{-/-}$ mutant mice
(N = 9, 9 weeks), heterozygous $Tbx15^{+/-}$ (N = 18, 9 weeks) and C57BL/6 $WT^{+/+}$ control mice
(N = 10, 9 weeks) were compared for ear and body morphological differences. (A) The schematic diagram of the one-step CRISPR/Cas9 technology used in *Tbx15* knockout mice. (B)
Example of left profile craniofacial photo of $Tbx15^{-/-}$ mutant mice with removal hair. (C) The
principal component analysis for 21 landmarks of $Tbx15^{-/-}$ mutant mice, heterozygous
$Tbx15^{+/-}$ and $WT^{+/+}$ mice. The upper layer shown the detailed contribution proportion of 21
landmarks to the first 10 principal components. The middle layer shown the screenplot of first
10 PCs, the significant association between the genotype and PCs which including PC1 and
PC4 (\* $P < 0.05$, \*\* $P < 0.01$, \*\*\* $P < 0.001$). The bottom shown the maximum PC1-, minimum PC1-,maximum PC4-, minimum PC4-, and mean ear shapes. (D) The pattern of genetic
association in humans (left) and in mice (right). (E) Effect of *Tbx15* knock-out on ear phenotypes in mice (blue for effect of heterozygote mutant and red for wildtype).
(TIF)

**S10 Fig. The *INTU* knockout mice using one-step CRISPR/Cas9 technology.** (A) The schematic diagram. (B) the sequence of gRNA. (C) Genotype was identified by PCR (Positive
251kb (loss-function), negative 402kb (wild type), heterozygous including 251kb and 402kb).
(TIF)

**S1 Table. Characteristics of 5 cohorts.**
(XLSX)

**S2 Table. Characteristics of 136 ear phenotypes in 5 cohorts.**
(XLSX)

**S3 Table. Kolmogorov-Smirnov normality tests of 136 ear phenotypes in 5 cohorts.**
(XLSX)

**S4 Table. The effects of age and sex on 136 ear phenotypes in the RS cohort.**
(XLSX)

**S5 Table. Replication of previously ear-associated SNPs in MinGWAS and C-GWAS.**
(XLSX)

**S6 Table. Replication of previously face-associated SNPs in MinGWAS and C-GWAS.**
(XLSX)

**S7 Table. GWAS Catalog entries for significant SNPs in MinGWAS and C-GWAS.**
(XLSX)

**S8 Table. Enrichment of 31 genes in the 16 ear-associated loci.**
(XLSX)

**S9 Table. The evidence of the 16 ear-associated lead SNPs or nearby genes in four databases including 3DSNP, VISTA, Harmonizome, and MGI.**
(XLSX)

**S10 Table. Partitioned heritability enrichments based on cell-type-specific regulatory
annotations.**
(XLSX)

**S11 Table. Effects of sex and genotypes on 28 ear phenotypes in mice.**
(XLSX)

**S12 Table. Definition of 17 human ear landmarks.**
(XLSX)

**S13 Table. Definition of 21 ear landmarks in mice.**
(XLSX)

**S14 Table. Enhancer regions estimated based on the H3K27ac and ATAC-seq using ROSE.**
(XLSX)

## Acknowledgments

The authors thank all sample donors for their contribution to this project. We would like to express our gratitude to Winston Rojas-Montoya for his invaluable support at the Universidad de Antioquia during the difficult period following the unfortunate passing of Gabriel Bedoya.

## Author Contributions

**Conceptualization:** Sijia Wang, Manfred Kayser, Fan Liu.

**Data curation:** Yi Li, Ziyi Xiong, Manfei Zhang, Pirro G. Hysi, Yu Qian, Kaustubh Adhikari, Jun Weng, Sijie Wu, Siyuan Du, Rolando Gonzalez-Jose, Lavinia Schuler-Faccini, Maria-Catira Bortolini, Victor Acuna-Alonzo, Samuel Canizales-Quinteros, Carla Gallo, Giovanni Poletti, Gabriel Bedoya, Francisco Rothhammer, Jiucun Wang, Jingze Tan, Ziyu Yuan, Li Jin, André G. Uitterlinden, Mohsen Ghanbari, M. Arfan Ikram, Tamar Nijsten, Xiangyu Zhu, Zhen Lei.

**Formal analysis:** Yi Li, Ziyi Xiong, Manfei Zhang, Pirro G. Hysi, Yu Qian, Kaustubh Adhikari.

**Funding acquisition:** Manfei Zhang, Li Jin, Peilin Jia, Andres Ruiz-Linares, Sijia Wang, Fan Liu.

**Investigation:** Yi Li, Yu Qian.

**Supervision:** Andres Ruiz-Linares, Timothy D. Spector, Sijia Wang, Manfred Kayser, Fan Liu.

**Visualization:** Yi Li, Ziyi Xiong.

**Writing – original draft:** Yi Li, Ziyi Xiong, Manfei Zhang, Manfred Kayser, Fan Liu.

**Writing – review & editing:** Yi Li, Ziyi Xiong, Manfei Zhang, Pirro G. Hysi, Yu Qian, Kaustubh Adhikari, Jun Weng, Sijie Wu, Siyuan Du, Rolando Gonzalez-Jose, Lavinia Schuler-Faccini, Maria-Catira Bortolini, Victor Acuna-Alonzo, Samuel Canizales-Quinteros, Carla Gallo, Giovanni Poletti, Gabriel Bedoya, Francisco Rothhammer, Jiucun Wang, Jingze Tan, Ziyu Yuan, Li Jin, André G. Uitterlinden, Mohsen Ghanbari, M. Arfan Ikram, Tamar Nijsten, Xiangyu Zhu, Zhen Lei, Peilin Jia, Andres Ruiz-Linares, Timothy D. Spector, Sijia Wang, Manfred Kayser, Fan Liu.

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
