## [Decision Letter · Decision Letter 0]

3 Jan 2023

Dear Dr Liu,

Thank you very much for submitting your Research Article entitled 'Combined multi-ethnic genome-wide association studies of 136 quantitative ear morphology traits identify 16 loci with 8 novel ones' to PLOS Genetics.

The manuscript was fully evaluated at the editorial level and by independent peer reviewers. The reviewers appreciated the attention to an important problem, but raised some substantial concerns about the current manuscript. Based on the reviews, we will not be able to accept this version of the manuscript, but we would be willing to review a much-revised version. We cannot, of course, promise publication at that time.

Should you decide to revise the manuscript for further consideration here, your revisions should address the specific points made by each reviewer. In particular, Reviewer 2 has made a number of suggestions that can lead to substantial improvement of the paper.  We will also require a detailed list of your responses to the review comments and a description of the changes you have made in the manuscript.  

If you decide to revise the manuscript for further consideration at PLOS Genetics, please aim to resubmit within the next 60 days, unless it will take extra time to address the concerns of the reviewers, in which case we would appreciate an expected resubmission date by email to plosgenetics@plos.org.

We are sorry that we cannot be more positive about your manuscript at this stage. Please do not hesitate to contact us if you have any concerns or questions.

Yours sincerely,

Hua Tang

Section Editor

PLOS Genetics

Scott Williams

Section Editor

PLOS Genetics

Reviewer's Responses to Questions

**Comments to the Authors:**

Reviewer #1: This is an impressive paper representing an enormous amount of work by many parties. The methods and results are interesting and convincing. Ideally there would be more detail but the paper is already long. My one request is that they should include a photo or a very good drawing to show what Darwin's tubercle is. My only other request is that the paper should be given a final read by a native English speaker since at the moment there are many spelling errors and infelicities of expression which detract from the overall quality.

Reviewer #2: In this manuscript, Li et al use quantitative phenotyping approaches as well as a new method for combining GWAS summary statistics from multiple traits to discover known and novel loci influencing variation in ear morphology in individuals of multiple ancestries. They provide evidence that candidate genes near these loci are preferentially expressed in cranial neural crest cells and having functions related to skeletal, craniofacial, and ear development. They show that one of the candidate genes, Intu, affects ear morphology upon heterozygous loss in mice.

Overall, this is a reasonably well-conducted study, with no fundamental claims that are unsupported by data. The deep learning phenotyping pipeline and C-GWAS methodological development could be potentially useful for others in the field. The measurement of ear morphology in knockout mice is another strength. One weakness is that there is a lack of independent replication, but the authors at least note this in their discussion. This is important information for readers to know when interpreting this manuscript. Another general point which confuses me is the authors’ use of the term “ectoderm-derived” when referring to traits such as ear morphology, face shape, etc. While it is true that many (although not all) of the contribution to these structures comes from ectoderm (i.e. CNCCs and surface ectoderm), there are numerous other tissues/traits that are also ectoderm-derived (it is, after all, one of the three germ layers) that presumably do not have the same genetics. If the authors are intending to make a more specific point about, for eg, surface ectoderm being a major contributor, then that should be stated, but in my opinion that is not well-supported here. It would be simpler to refer to these related traits as “craniofacial traits” or something along those lines.

There are several claims made in the manuscript which in my opinion should be adjusted or supported with additional analyses:

Line 85, it is somewhat of an exaggeration to say that this study “corrects the long-term view that ear phenotypes are genetically simple traits.” Indeed previous ear morphology GWAS had already found 8 loci, and combined with the fact that related traits such as face and cranial shape have hundreds of associated loci and are clearly genetically complex, there is very little evidence for such a view of ear morphology being simple.

The authors claim (lines 148-149) that their deep learning phenotyping pipeline “convincingly” outperforms human perception in ear landmarking. This is not well supported by the data. The left-right ear correlation (which the authors consider a gold standard of accuracy) for deep learning is 0.52 with sd 0.1, while the correlation for human raters is 0.44 with sd 0.2. Thus human raters are well within one sd with respect to this measure of accuracy. All that can be said from this is that the performance of the deep learning pipeline is roughly similar to that of human raters. This is not a problem (there is no need to demonstrate superior accuracy) but this claim should be rephrased accordingly.

The expression analysis of genes near the 16 loci is not convincing. An appropriate background set would be genes near to (or closest to) a random set of control, frequency-matched SNPs that are the same size as the true ear morphology-associated SNPs. Such sets can easily be obtained using a tool such as SNPsnap. Expression within CNCCs, as well as comparing CNCCs to other tissues, should be performed with such a background set. The authors should also perform heritability enrichment analysis (i.e. stratified LD score regression) for the univariate ear morphology traits (only TwinsUK perhaps) with respect to active regulatory regions from CNCCs and other cell types, all of which are publicly available. This would be much more convincing when claiming the importance of certain cell types for ear morphology and would avoid the issue of analyzing small sets of genes/SNPs for statistical significance.

Some of the analyses of ear morphology, in particular comparing PC loadings between genotypes, is not entirely statistically sound. Comparing PC loadings between samples upon which PC was initially performed will exaggerate statistical significance (and indeed even with this PC2 is only weakly associated with genotype). I believe that there are significant (albeit weak) differences in ear morphology between Intu hets and WT, as indicated by the univariate landmark analysis in 3F. If the authors wish to perform an integrated morphological analysis by genotype, they should use a generalized Procrustes analysis framework (such as the one available in the R package geomorph), which would allow them to test for overall shape changes as a function of genotype. The effects of Tbx15 KO are stronger, but the manuscript would benefit from similar analyses for Tbx15 het/KO mice as well.

**Have all data underlying the figures and results presented in the manuscript been provided?**

Reviewer #1: Yes

Reviewer #2: Yes

PLOS authors have the option to publish the peer review history of their article (what does this mean?). If published, this will include your full peer review and any attached files.

Reviewer #1: No

Reviewer #2: **Yes: **Sahin Naqvi

---

## [Decision Letter · Decision Letter 1]

4 Apr 2023

Dear Dr Liu,

Thank you very much for submitting your Research Article entitled 'Combined genome-wide association study of 136 quantitative ear morphology traits in multiple populations reveal 8 novel loci' to PLOS Genetics.

The manuscript was fully evaluated at the editorial level and by independent peer reviewers. The reviewers appreciated the attention to an important topic but identified some concerns that we ask you address in a revised manuscript.

We therefore ask you to modify the manuscript according to the review recommendations. Your revisions should address the specific points made by each reviewer.

Yours sincerely,

Hua Tang

Section Editor

PLOS Genetics

Scott Williams

Section Editor

PLOS Genetics

Reviewer's Responses to Questions

**Comments to the Authors:**

Reviewer #1: the authors have satisfied all my concerns and the paper is now fit to be published

Reviewer #2: The authors have addressed all of my comments, except for one important error in which they refer to embryonic craniofacial tissue samples (CS13, 14, 15, 17 in the S-LDSC analysis in Fig 2c) as CNCCs. These are not CNCCs, they are primary tissue from embryonic craniofacial regions which will be a mix of many cell types. More details are found in the cited paper (Wilderman et al 2018). For CNCCs the authors should analyze ATAC-seq/H3K27ac peaks from hESC-derived CNCCs, available from Prescott et al (ref 61 in the paper).

**Have all data underlying the figures and results presented in the manuscript been provided?**

Reviewer #1: Yes

Reviewer #2: Yes

PLOS authors have the option to publish the peer review history of their article (what does this mean?). If published, this will include your full peer review and any attached files.

Reviewer #1: No

Reviewer #2: **Yes: **Sahin Naqvi

---

## [Decision Letter · Decision Letter 2]

16 May 2023

Dear Dr Liu,

We are pleased to inform you that your manuscript entitled "Combined genome-wide association study of 136 quantitative ear morphology traits in multiple populations reveal 8 novel loci" has been editorially accepted for publication in PLOS Genetics. Congratulations!

Yours sincerely,

Hua Tang

Section Editor

PLOS Genetics

Scott Williams

Section Editor

PLOS Genetics

Comments from the reviewers (if applicable):

Reviewer's Responses to Questions

**Comments to the Authors:**

Reviewer #2: I am satisfied with the changes authors have made

**Have all data underlying the figures and results presented in the manuscript been provided?**

Reviewer #2: None

PLOS authors have the option to publish the peer review history of their article (what does this mean?). If published, this will include your full peer review and any attached files.

Reviewer #2: **Yes: **Sahin Naqvi

**Data Deposition**

http://datadryad.org/submit?journalID=pgenetics&manu=PGENETICS-D-22-01067R2

**Press Queries**

---

## [Editor Report · Acceptance letter]

15 Jun 2023

PGENETICS-D-22-01067R2 

Combined genome-wide association study of 136 quantitative ear morphology traits in multiple populations reveal 8 novel loci 

Dear Dr Liu, 

We are pleased to inform you that your manuscript entitled "Combined genome-wide association study of 136 quantitative ear morphology traits in multiple populations reveal 8 novel loci" has been formally accepted for publication in PLOS Genetics! Your manuscript is now with our production department and you will be notified of the publication date in due course.

With kind regards,

Bernadett Koltai

PLOS Genetics

On behalf of:
